# The triangulation of multimorbidity: A systematic review of primary sleep disorders, hypertension, and psychiatric disorders

Chadia Haddad [1,2,3,4*], Hala Sacre[2], Samah Tawil [1,2*], Pascale Salameh[1,2,5,6☺], Sola Aoun Bahous [1,7☺]

**1** Gilbert and Rose-Marie Chagoury, School of Medicine, Lebanese American University, Beirut, Lebanon, **2** INSPECT-LB (Institut National de Santé Publique, d'Épidémiologie Clinique et de Toxicologie-Liban), Beirut, Lebanon, **3** Research Department, Psychiatric Hospital of the Cross, Jal Eddib, Lebanon, **4** Faculty of Public Health, Lebanese University, Fanar, Lebanon, **5** Department of Primary Care and Population Health, University of Nicosia Medical School, Nicosia, Cyprus, **6** Faculty of Pharmacy, Lebanese University, Hadat, Lebanon, **7** Lebanese American University Medical Center – Rizk Hospital, Beirut, Lebanon

☺ These authors share senior authorship.
\* chadia_9@hotmail.com (CH); samah.tawil@lau.edu.lb (ST)

## Abstract

Despite the growing evidence of the interconnectedness of hypertension, sleep disorders, and mental health, the exact nature of the relationships and the potential for combined or synergistic effects remain unclear. Potential mechanisms include environmental factors, family stressors, financial difficulties, treatment side effects and shared pathophysiological comorbidities. Therefore, this systematic review aimed to address this gap by assessing comprehensively the interrelationships between these three conditions among adults. A systematic review was conducted in line with the preferred reporting items for systematic reviews and meta-analyses. The literature search was performed across three databases: PubMed, SCOPUS, and CINAHL. From an initial pool of 1759 articles, 45 met the inclusion criteria and were used in the analysis. Most studies assessed the associations between the three conditions pairwise, using different definitions and methods. Positive (60–75%) or non-significant associations (25–40%) were commonly reported, with no inverse associations identified. This consistent pattern suggests that these conditions are interrelated, even when statistical significance was not reached in some cases. In conclusion, the association between sleep disorders, hypertension, and psychiatric diseases is highlighted in the majority of studies, showing predominantly positive or non-significant relationships, with no studies reporting inverse associations among these three conditions. These findings suggest that addressing these conditions in an integrated manner may improve clinical management and patient outcomes.

**Data availability statement:** All relevant data are within the paper and its Supporting information files.

**Funding:** The authors received no specific funding for this work.

**Competing interests:** The authors have declared that no competing interests exist.

## Introduction

According to DSM-V criteria, sleep disorders include insomnia and obstructive sleep apnea (OSA), as well as hypersomnolence disorder, narcolepsy, breathing-related sleep disorders, circadian rhythm sleep-wake disorders, non-rapid eye movement (NREM), sleep arousal disorders, nightmare disorder, rapid eye movement (REM), sleep behavior disorder, restless legs syndrome, and substance/medication-induced sleep disorders. Sleep disorders have emerged as significant concerns affecting millions of adults worldwide. These disorders have been associated with various mental and physical health problems, contributing to an increased risk of cardiovascular diseases, including hypertension, and psychiatric disorders such as depression and anxiety [1,2]. Globally, the prevalence of hypertension in adults is estimated to be around 33% [3], while up to 20% of adults suffer from some type of sleep disorder [4], and nearly 5% are diagnosed with depression [5].

Insomnia, hypertension, and common mental health disorders, such as anxiety and depression, are leading contributors to the global burden of disease and are connected through complex, bidirectional relationships [6–9]. Their frequent co-occurrence poses a significant public health concern due to their effects on healthcare utilization, job productivity, and quality of life [10,11].

Sleep deprivation and sleep-related disorders alone affect nearly all primary health indicators, including mortality, morbidity, performance, accidents and injuries, family well-being, and daily functioning. Some consequences, such as motor vehicle accidents, can occur within hours of sleep disruption, underscoring the urgency of addressing sleep issues [12].

Hypertension remains one of the most pressing global health challenges, contributing to approximately eight million deaths annually and affecting two in five individuals in the Eastern Mediterranean Region [13]. In addition to being one of the most prevalent yet modifiable risk factors, hypertension significantly increases the risk of peripheral vascular disease, congestive heart failure, coronary heart disease, stroke, and chronic kidney disease [14].

Mental health issues are fast becoming one of society's most significant health challenges. Beyond increasing pressure on psychiatric services, poor mental health disrupts interpersonal relationships, fosters social isolation, and imposes emotional and financial burdens on families. Left unaddressed, it may lead to long-term behavioral and social difficulties [15].

Research has shown that short sleep duration and insomnia contribute to hypertension, potentially through mechanisms like increased inflammation, dysregulation of the hypothalamic-pituitary-adrenal axis, and sympathetic nervous system activation [16,17]. Similarly, OSA has been associated with an increased risk of incident hypertension, likely due to intermittent hypoxia, oxidative stress, and endothelial dysfunction [18,19]. Furthermore, insomnia is a well-established risk factor for incident depression [20,21] and anxiety [22,23], possibly due to polymorphisms in the circadian clock genes and shared neurobiological pathways involving serotonin, dopamine, and the stress response system [24]. Insomnia has also been associated with

increased physical health problems, including hypertension [25]. Its impact extends beyond physical health, with different phenotypes presenting varying risks to mental health [26,27]. Additionally, mood disorders, such as depression, may lead to various functional physical disorders, including uncontrolled hypertension [28].

While previous systematic reviews and meta-analyses have explored the associations between hypertension and insomnia [29,30], the role of mental health disorders in this context has often been overlooked. Few studies have comprehensively examined the prevalence and impact of sleep disorders in people with both hypertension and psychiatric disorders [31,32]. Moreover, the mediating role of insomnia has been explored between post-traumatic stress disorder (PTSD) and hypertension [33] and between hypertension and depression [34]. However, methodological differences across these studies, including variations in how insomnia and psychiatric disorders were defined and measured and the absence of systematic reviews on this topic, have contributed to a fragmented knowledge of this triangular relationship. Insomnia, hypertension, and anxiety/depression are interconnected, and understanding their interaction is crucial for improving clinical management and treatment outcomes, reducing healthcare costs, and enhancing patients' quality of life. Despite the growing evidence of their interconnectedness, the exact nature of the relationships and the potential for combined or synergistic effects remain unclear.

In clinical settings, insomnia, hypertension, and anxiety or depression might co-occur, complicating diagnosis and treatments and influencing one another bidirectionally; for example, insomnia can exacerbate hypertension and mood disorders, while anxiety and depression can worsen sleep disturbances, resulting in a vicious cycle. Given their global prevalence, frequent comorbidity, and significant individual and societal impact, research into the relationships between these conditions is essential, as they are associated with increased mortality, disability, healthcare expenses, and a substantial share of the global disease burden [35].

This systematic review aims to assess the interrelationships between sleep disturbances, hypertension, and psychiatric disorders in adults. By identifying potential shared clinical and behavioral pathways, it seeks to inform more integrated and patient-centered approaches to care—beyond traditional biopsychosocial or purely biological frameworks.

## Materials and methods

### Protocol and registration

This systematic review was conducted in accordance with the Preferred Reporting Items for Systematic Reviews and Meta-Analyses (PRISMA) guidelines (S1 PRISMA Checklist) and registered in the International Prospective Register of Systematic Reviews (PROSPERO, protocol ID: CRD42024595467).

### Search strategy

In line with the preferred reporting items for systematic reviews and meta-analyses (PRISMA) guidelines [36], an electronic systematic search of PubMed, SCOPUS, and CINHAL was performed to identify relevant observational studies and clinical trials published in English between January 2012 and December 2023, examining the associations between sleep problems (including insomnia, sleep disturbances, and sleep disorders), hypertension, and psychiatric disorders. Studies examining at least two of the three conditions of interest (sleep disorders, hypertension, or psychiatric disorders) in adult populations were included.

Medical subject headings (MeSH) and keywords included "sleep initiation and maintenance disorders"[Mesh] OR sleep initiation and maintenance disorders[tiab] OR disorders of initiating and maintaining sleep[tiab] OR DIMS[tiab] OR early awakening[tiab] OR awakening, early[tiab] OR insomnia[tiab] OR insomnias[tiab] OR sleep initiation dysfunction[tiab] OR sleep initiation dysfunctions[tiab] OR dysfunction, sleep initiation[tiab] OR dysfunctions, sleep initiation[tiab] OR sleeplessness[tiab] OR sleep disturbance[tiab] OR sleep disorder[tiab] OR altered sleeping pattern[tiab] AND Hypertension[mesh] OR hypertension[tiab] OR blood pressure, high[tiab] OR blood pressures, high[tiab] OR high blood pressure[tiab] OR high

blood pressures[tiab] "Diastolic Pressure" OR "Pulse Pressure" OR "Systolic Pressure" OR "Blood Pressure, High" AND "mental disorders" OR "psychotropic drugs"[Mesh] OR drugs, psychotropic[tiab] OR drug, psychotropic[tiab] OR psycho-active agent[tiab] OR agents, psychoactive[tiab] OR agent, psychoactive[tiab] OR psychoactive agents[tiab] OR psycho-active drug[tiab] OR psychoactive drugs[tiab] OR drug, psychoactive[tiab] OR drugs, psychoactive[tiab] OR psychotropic drug[tiab] OR psychopharmaceutical[tiab] OR psychopharmaceuticals[tiab].

## Inclusion and exclusion criteria

This systematic review considered studies that included an adult human sample aged 18 and older, with abstracts and full-text papers published in English between January 2012 and December 2023. Eligible study designs encompassed cross-sectional, longitudinal, prospective, retrospective, interventional, and experimental studies, including randomized controlled trials (RCTs). Studies were excluded if they were case studies, commentaries, editorials, or letters, or if the full text was unavailable. Meta-analyses and review articles were also excluded but screened for additional relevant studies not found in the initial electronic search.

Inclusion criteria were based on three assessments:

1.  Hypertension: studies involving patients with self-reported or diagnosed hypertension or blood pressure >140/90 mmHg were included.

2. Sleep disorders: studies were included if they used sleep questionnaires for insomnia, oversleeping, or any sleep disturbance based on DSM-IV diagnostic criteria. These studies evaluated patient symptoms, such as difficulties falling asleep, waking up several times a night, and morning fatigue, or employed any sleep assessment scale like the Insomnia Severity Index (ISI) or Pittsburgh Sleep Quality Index (PSQI).

3. Psychiatric disorders: studies were included if they involved any psychiatric disorder (e.g., anxiety, depression, psychosis, obsessive-compulsive disorder, or PTSD), utilized physician diagnoses or diagnostic tools (such as Beck Depression Inventory, Beck Anxiety Inventory, General Anxiety Disorder, PTSD Checklist, Center for Epidemiologic Studies Depression Scale), or reported the use of anti-psychotics, anti-depressants, sedatives, or neuroleptics.

## Data review and extraction

Three authors independently evaluated eligibility and extracted the data. A data extraction template was created in Microsoft Excel to coordinate article screening and remove duplicate entries, errata, and corrections. In a two-stage process, two coders first reviewed article abstracts and then full-text manuscripts for eligibility. They also completed a data extraction template for each article screened for inclusion. Discrepancies were resolved through discussion between the authors until a consensus was reached. This template included fields for (i) article title, (ii) author names, (iii) author count, (iv) citations count, (v) journal name, (vi) study location, design, target population, sample size, and sample characteristics (e.g., mean age, age range, gender distribution), (vii) description of each disease (type, diagnostic tests, symptoms, and percentages of affected people), (vii) main findings to evaluate the relationship results between two variables at a time then the relationship among all variables (if feasible) [e.g., odds ratio (OR), relative risk (RR), prevalence ratios (PR), or hazard ratios (HR) with 95% confidence intervals (CI) and p-values when provided], (viii) covariates (e.g., demographics, co-morbidity), and (viii) keywords. The follow-up period was also extracted for longitudinal studies.

## Quality assessment

Quality assessments were completed using an adapted version of widely used scales, i.e., the Risk of Bias In Non-randomized Studies – of Exposures (ROBINS-E) assessment tool for observational studies (S1 File) [37] and the Revised Cochrane risk-of-bias tool for cluster-randomized trials (RoB 2 CRT) for experimental studies [38]. Items were designed

to evaluate the methodology of each article, including its research integrity, results, and relevance. Several questions assessed the quality of the study (e.g., recruitment of participants, deviations from intended interventions, missing outcome data, measurement of outcome, selection of reported results) with lower scores indicating poorer quality and higher scores indicating greater quality [37,38].

## Results

### Literature search results

The database search yielded 1759 publications (S1 Data): 559 from PubMed, 1160 from Scopus, and 40 from CINAHL. After initial screening, 1697 articles were excluded due to duplication (n = 77) and irrelevance (n = 1620). The remaining 62 articles underwent abstract assessment, resulting in the exclusion of seven more articles (1 duplicate, 4 irrelevant topics, and 2 review articles). The full texts of the 55 retained articles were examined, and six additional articles were included after reference screening. From these 61 articles, 16 were excluded due to duplication (n = 1), non-English language (n = 1), irrelevant topics (n = 8), RCT protocol (n = 1), not including HTN as a main variable (n = 4), and involving participants under 18 years. Ultimately, 45 articles met the inclusion criteria and were selected for review (Fig 1).

### Studies' characteristics

The 45 publications differed substantially in their design, sample size, location, and type of population. These studies originated from different regions: Middle East and North Africa (MENA) (n = 2), Sub-Saharan Africa (n = 3), South Asia (n = 3), East, Southeast Asia, & Pacific (n = 10), Northern & Western Europe (n = 13), Central and Eastern Europe and Central Asia (n = 1), North America (n = 12), and Latin America and the Caribbean (LAC) (n = 1). Half of the studies were cross-sectional (n = 23), 20 were cohorts, one study was a case-control, and one was a randomized control trial.

Sample sizes ranged from 19 to 62,253,910 adult participants, with most studies involving healthy populations (n = 18). Some studies focused on specific groups, such as patients with hypertension (n = 4), cardiovascular disease (n = 4), psychiatric conditions (n = 1), depression (n = 3), sleep disorders (n = 2), epilepsy (n = 1), Parkinson's disease (n = 1), veterans (n = 4), and postmenopausal women (n = 1). Additionally, seven studies involved older adults. The majority of the included studies relied on self-administered questionnaires or face-to-face clinical interviews for data collection.

### Outcome results

Tables 1–6 describe the associations between sleep problems, psychiatric disorders, and hypertension based on the findings of each study. Psychiatric disorders were predominantly represented by depression and anxiety disorders. As for sleep problems, insomnia was frequently observed, alongside other sleep complaints such as difficulty falling asleep and sleep duration. Hypertension was also measured using different criteria across studies. The associations between dependent and independent variables were typically presented as odds ratios (OR), hazard ratios (HR), and beta coefficients, depending on the study design and whether the variables were dichotomous or continuous. Thus, conducting a meta-analysis of these associations was not feasible due to discrepancies in the definitions and methods employed across studies. Consequently, the results were presented narratively, with the number of positive or negative associations counted within each study.

### Studies demonstrating three sets of two-by-two associations

No studies investigated the interaction between the three disorders at the same time, except for two studies where the mediating role of insomnia has been explored between PTSD and hypertension [33] and between hypertension and depression [34].

Global Public Health

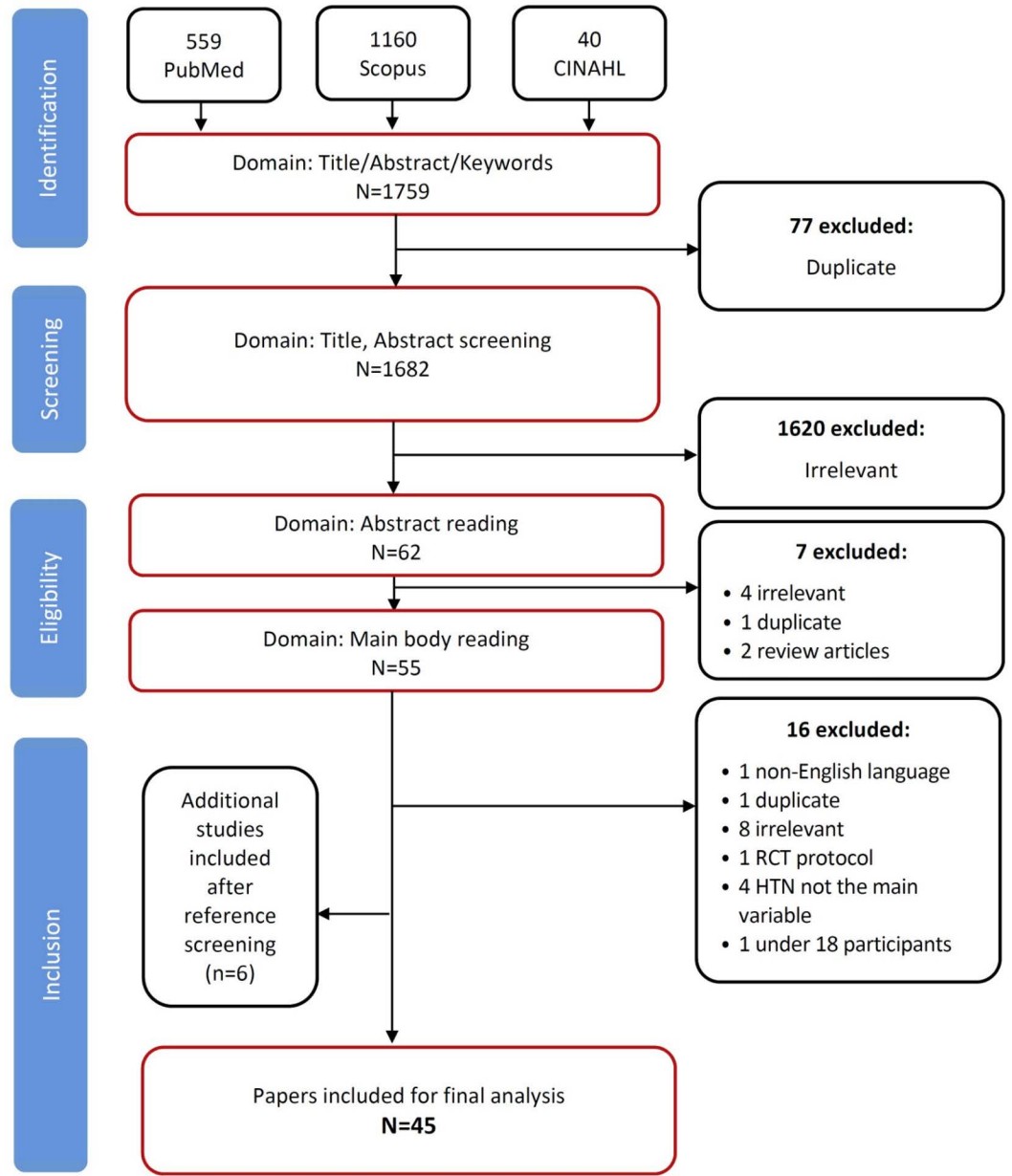

**Fig 1. PRISMA flow diagram of study selection.**

When considering the three types of two-by-two associations (sleep problems and psychiatric disorders [S-P], sleep problems and HTN [S-H], and psychiatric disorders and HTN [P-H]), four studies were identified (Table 1). The findings were inconsistent due to methodology differences, even among studies with large sample sizes.

For the psychiatric disorder assessment, validated scales, such as the Center for Epidemiological Studies-Depression (CES-D) scale and the Hamilton Depression Rating Scale (HAM-D), were mainly used to diagnose depression. PTSD, when included as a psychiatric variable, was assessed using the PTSD checklist-Civilian version (PCL-C) scale [33]. Sleep problems were evaluated through various methods: one study used a self-reported question on insomnia [32], while

**Table 1. Description and results of the studies demonstrating three types of association, the association between sleep problems and psychiatric disorders, association between sleep problems and HTN and the association between psychiatric disorders and HTN.**

| Study characteristics | | | Measurement tools | | | Outcome results | | | | | | |
|---|---|---|---|---|---|---|---|---|---|---|---|
| | | | | | | Association between sleep problems and psychiatric disorders [S-P] | | Association between sleep problems and HTN [S-H] | | Association between HTN and psychiatric [P-H] | | |
| Author (year) | Risk of Bias | Study design (sample size, population type, location) | Psychiatric disorders | Sleep problems | Hypertension | Variables | results | Variables | results | Variables | results |
| Poole L et al. (2018) [39] | Low | Longitudinal Study (N=5172; Elderly; England) | Epidemiological Studies Depression scale (CES-D) | Three questions derived from the Jenkins Sleep Problems Scale | Self-reported (yes/no) and objective assessments (hypertension defined as systolic blood pressure >140 and diastolic blood pressure >90) | DV: depression IV: sleep complaints | Beta=0.067 95% CI: 0.135–0.311, p<0.001 (positive association) | DV: Sleep complaints IV: HTN | OR = 0.97 95% CI: 0.84–1.14, p=0.753 (no association) | DV: depression IV: HTN | Beta=-0.006, 95% CI -0.098-0.062, p=0.659 (no association) |
| Dong Y et al (2019) [32] | Low | Longitudinal study (N=18123; Elderly; USA) | CES-D scale | Self-reported four questions about insomnia | Self-reported hypertension asking a yes/no response question | DV: Depression IV: insomnia | Insomnia symptoms were predictors of depression by 9.0 to 11.5 folds, depending on age group (positive association) | DV: HTN IV: insomnia | Insomnia symptoms were significant predictors of hypertension (positive association) | DV: depression IV: blood pressure | SBP, DBP, were not significant predictors of depression (p>0.05) (no association) |
| Ma L (2017) [40] | Some concerns | Case-control (N=73; Elderly, China) | HAM-D questionnaire | Sleep quality (PSQI scale) | Blood pressure readings were taken from the right arm after 5 min of physical inactivity. Systolic and diastolic blood pressures were measured by Korotkoff I and V. | DV: depression IV: PSQI | r=0.432, P<0.01 (positive association) | control group: hypertension without depression case group: hypertension with depression | Subjective sleep quality, time to sleep, sleep duration, sleep efficiency, sleep disorders and anti-insomnia drugs were worse in the case group. p<0.005 (positive association) | control group: hypertension without depression case group: hypertension with depression | The case group had higher 24 h ambulatory SBP and DBP, and nocturnal SBP and DBP than the control group, p<0.005 (positive association) |
| Gaffey AE et al (2020) [33] | Some concerns | Cross-sectional (N=1109; Women veterans, USA) | PTSD checklist – Civilian version (PCL-C) | The Insomnia Severity Index (ISI) | One question, 'In the past 12 months, have you received medical treatment for high blood pressure?' Confirmatory responses were classified as hypertension diagnosis. | DV: Insomnia IV: PTSD | Beta=0.73 95% CI: 0.69–0.77, p<0.001 (Positive association) | DV: HTN IV: insomnia | Beta=0.12 95% CI 0.02; 0.23, p=0.02 (Positive association) | DV: HTN IV: PTSD | Beta=0.02, 95%CI -0.08, 0.12, p=0.66 (no association) |

DV: dependent variable; IV: independent variable; HTN: hypertension; OR: odds ratio, CI: Confidence interval; HRa: adjusted Hazard ration, RR: risk ratio; HBP: high blood pressure; CCB: Calcium channel blockers; CVD: cardiovascular disease; MS: metabolic syndrome.

Global Public Health
PLOS

**Table 2. Description and results of the studies demonstrating two types of association, the association between sleep problems and psychiatric disorders and the association between psychiatric disorders and HTN.**

| Author (year) | Risk of Bias | Study design (sample size, population type, location) | Measurement tools | | | Outcome results | | | | | |
|---|---|---|---|---|---|---|---|---|---|---|---|
| | | | Psychiatric disorders | Sleep problems | Hypertension | Association between sleep problems and psychiatric disorders [S-P] | | | Association between psychiatric disorders and Hypertension [P-H] | | |
| | | | | | | Variables | Results | Types of association | Variables | Results | Types of association |
| Babu AR et al. (2019) [41] | Some concerns | cross-sectional study (N = 1210; Adult women; India) | GAD-7 | Dichotomic question (Yes/No) | Dichotomic question (Yes/No) | DV: Anxiety IV: Sleep problems | OR = 1.64; 95% CI 1.77 – 2.91, p = 0.004 | Positive | DV: Anxiety IV: HTN | OR = 1.83; 95% CI 1.04 – 3.23, p = 0.034 | Positive |
| Son YS et al. (2018) [42] | Some concerns | cross-sectional study (N = 846; patients with hypertension; Korea) | Patient Health Questionnaire (PHQ-9) | self-reported response to the following question: "How many hours do you sleep per night on average? | A device that assesses blood pressure | DV: depression IV: Sleep duration | OR = 4.09; 95% CI 1.83 – 9.13, p < 0.05 | Positive | DV: Depressive symptoms IV: Blood pressure | Mean SBP: 128.02 ± 15.76 vs 126.04 ± 15.24, p = 0.259 Mean DBP: 75.4 4 ± 10.40 vs 73.82 ± 9.87, p = 0.157 | No association |
| Nasir et al. (2015) [43] | Some concerns | cross-sectional study (N = 40; Patients with congestive heart failure; Pakistan | Beck Depression Inventory | PSQI | presence of HTN was collected from the medical record | DV: severity of depression IV: Insomnia | r = 0.187; p = 0.005 | Positive | DV: depression IV: HTN | r = 0.313; p = 0.049 | Positive |
| Meira B et al (2022) [44] | Low | Longitudinal Study (N = 73; Parkinson; England) | Parkinson psychosis questionnaire | Polysomnography [PSG] | blood pressure values >140/90 mm Hg | DV: Parkinson IV: OSA | Beta = 2.11 95% CI: -0.08– 4.30, p > 0.05 | (No association) | DV: Parkinson IV: HBP | Beta = 0.26, 95% CI 1.344; 1.872, p = 0.744 | (no association) |

DV: dependent variable, IV: independent variable, OR: Odds Ratio, CI: confidence interval, HTN: hypertension, PSQI: Pittsburgh Sleep Quality Index.

the others employed different tools, i.e., the Jenkins Sleep Scale (JSS), the Pittsburgh Sleep Quality Index (PSQI), and the Insomnia Severity Index (ISI) [33,39,40]. Three studies relied on self-reported data for hypertension [32,33,39], while one used clinical devices for blood pressure measurement [40].

In studies examining sleep problems and depression [S-P], all four [32,33,39,40] found a positive association between these two conditions. Two had a low risk of bias [32,39] and focused on older adults, while the other two [33,40], which included both elderly and women veterans, had some bias concerns. For the association between sleep disorders and hypertension [S-H], three studies [32,33,40] showed a positive association, and one reported no link between the two conditions despite its large sample size, low risk of bias, and use of validated measurement tools [39]. For depression and HTN [P-H], one study [40] found a positive association, while the three others [32,33,39] reported no association (Table 1).

**Table 3. Description and results of the studies demonstrating two types of association, the association between sleep problems and psychiatric disorders and the association between sleep problems and HTN.**

| Author (year) | Risk of Bias | Study design (sample size, population type, location) | Measurement tools | | | Outcome results | | | | | |
|---|---|---|---|---|---|---|---|---|---|---|---|
| | | | Psychiatric disorders | Sleep problems | Hypertension | Association between sleep problems and psychiatric disorders [S-P] | | | Association between sleep problems and HTN [S-H] | | |
| | | | | | | Variables | Results | Types of association | Variables | Results | Types of association |
| Basnet S et al. (2016) [54] | Low | cross-sectional study (6424; Adults, Finland) | One question (Yes/No) | One question (Yes/No) | One question (Yes/No) | DV: depression IV: sleep quality | OR = 1.89; 95% CI 1.34 – 2.68, p<0.05 | Positive | DV: HTN IV: Sleep quality | OR = 1.18; 95% CI 0.87 – 1.59, p>0.05 | No association |
| Kim JM et al. (2017) [45] | Low | longitudinal study (N=1152; Patients with acute coronary syndrome; Korea) | MINI | Leeds Sleep Evaluation Questionnaire (LSEQ) | self-reported diagnoses of HTN | DV: depressive disorders IV: quality of sleep | OR = 1.66; 95% CI 1.00 – 2.75, p<0.05 | Positive | DV: HTN IV: sleep disturbance | The sleep disturbance factors were significantly associated with hypertension (p<0.05) | Positive |
| Wang S. et al. (2017) [46] | Low | cross-sectional study (N=4115; Older participants; China) | 12-item General Health Questionnaire (GHQ-12) | One question: How many hours do you sleep each day (24h) on average in the past month? | 10th Revision (ICD-10) | DV: sleep duration IV: mental health | OR = 1.29; 95% CI 1.01 – 1.64, p<0.05 | Positive | DV: sleep duration IV: HTN | OR = 1.36; 95% CI 1.09 – 1.70, p<0.05 | Positive |
| Seow LS et al. (2020) [55] | Low | cross-sectional study (N=6126; Adults; Singapore) | WMH-CIDI version 3.0 (based on DSM-IV; ICD-10) | PSQI scale | Modified version of the CIDI 3.0 checklist | DV: MDD IV: sleep quality | OR = 2.00; 95% CI 1.40 – 2.90, p<0.05 | Positive | DV: sleep quality IV: HTN | OR = 1.08; 95% CI 0.84 – 1.40, p=0.523 | No association |
| Ayanaw T et al. (2022) [47] | Low | cross-sectional study (N=563; Hypertensive adults; Ethiopia) | Hospital Anxiety and Depression Scale | PSQI scale | Diagnosed as HTN while attending a chronic follow-up care clinic | DV: sleep quality IV: depression | OR = 2.03; 95% CI 1.23 – 3.34, p<0.05 | Positive | DV: sleep quality IV: HTN | OR = 1.78; 95% CI 1.01 – 3.12, p<0.05 | Positive |
| Laaboub N et al. (2022) [48] | Low | longitudinal cohort (N=2861; Psychiatric patients; Swiss) | ICD-10 classification | ICD 10 "F51.0" diagnosis | systolic BP ≥130 or diastolic BP ≥85mm Hg or treatment for hypertension | DV: insomnia IV: psychiatric diagnosis | Insomnia disorders observed in schizoaffective and bipolar disorders (13% vs 8% and 20% vs 17%, p<0.05) | Positive | DV: insomnia IV: HTN | OR = 1.86; 95% CI 1.23 – 2.81, p<0.05 | Positive |

*(Continued)*

**Table 3.** (Continued)

| Author (year) | Risk of Bias | Study design (sample size, population type, location) | Measurement tools | | | Outcome results | | | | | |
|---|---|---|---|---|---|---|---|---|---|---|---|
| | | | Pyschiatric disorders | Sleep problems | Hypertension | Association between sleep problems and psychiatric disorders [S-P] | | | Association between sleep problems and HTN [S-H] | | |
| | | | | | | Variables | Results | Types of association | Variables | Results | Types of association |
| Sivertsen B et al (2014) [56] | Low | prospective longitudinal design (N=24715; Adults; Norway) | Hospital Anxiety and Depression Scale (HADS) | DSM-IV diagnosis | Self-reported diagnoses | DV: insomnia IV: depression | OR = 2.68; 95% CI 2.28 – 3.14, p<0.05 | Positive | DV: insomnia IV: HTN | OR = 1.04; 95% CI 0.92 – 1.18, p>0.05 | No association |
| Falkingham J. (2022) [49] | Low | prospective longitudinal design (N=12804; Participants aged 16 and above; UK) | Self-reported diagnoses | PSQI | Self-reported diagnoses | DV: sleep problems IV: emotional, nervous or psychiatric problem | OR = 5.01; 95% CI 3.02 – 8.34, p<0.05 | Positive | DV: sleep problems IV: HTN | OR = 2.54; 95% CI 1.51 – 4.26, p<0.05 | Positive |
| Muhammad T et al. (2021) [50] | Low | longitudinal study (N=31358; Older adults; India) | One question (Yes/No) | Series of questions about insomnia | Self-reported diagnoses of HTN (Yes/No) | DV: insomnia IV: psychiatric disorders | OR = 1.75; 95% CI 1.48 – 2.06, p<0.05 | Positive | DV: insomnia IV: HTN | OR = 1.35; 95% CI 1.27 – 1.43, p<0.05 | Positive |
| Ulmer CS et al. (2015) [57] | Low | cross sectional study (N=1855; Military personnel and Veterans; USA) | Beck Depression Inventory-II | Sleep difficulties were assessed using 2 items from the Symptom Checklist-90-Revised | Self-reported diagnoses of HTN (Yes/No) | DV: sleep problems IV: depression | OR = 4.41; 95% CI 2.87 – 6.78, p<0.05 | Positive | DV: sleep difficulties IV: HTN | OR = 1.39; 95% CI 0.91 – 2.14, p>0.05 | No association |
| Gupta et al. (2014) [58] | Low | retrospective case-control study (N=62,253,910; OSA patients versus controls; USA) | ICD9-CM | ICD9-CM | ICD9-CM | DV: Sleep apnea IV: depressive disorders | OR = 0.81; 95% CI 0.26 – 2.51, p>0.05 | No association | DV: sleep apnea IV: HTN | OR = 1.83; 95% CI 1.27 – 2.65, p=0.001 | Positive |
| Johann AF et al. (2017) [52] | Low | Retrospective case-control study (N=328; Adults; Germany) | Beck depression inventory | Polysomnography | boso medicus uno device (HTN was defined SBP ≥140mm Hg, a DBP ≥90mm Hg | DV: sleep duration IV: depression | Mean BDI in short sleep =6.1 vs 6.6 in normal sleep | No association | DV: sleep duration IV: HTN | OR = 0.80; 95% CI 0.41 – 1.55, p>0.05 | No association |

*(Continued)*

**Table 3.** (Continued)

| Author (year) | Risk of Bias | Study design (sample size, population type, location) | Measurement tools | | | Outcome results | | | | | |
|---|---|---|---|---|---|---|---|---|---|---|---|
| | | | Pyschiatric disorders | Sleep problems | Hypertension | Association between sleep problems and psychiatric disorders [S-P] | | | Association between sleep problems and HTN [S-H] | | |
| | | | | | | Variables | Results | Types of association | Variables | Results | Types of association |
| Routledge FS et al. (2017) [59] | Low | Longitudinal cohort (N=496; adults; Georgia) | BDI-II | PSQI | BP monitor | DV: Insomnia IV: depression | $M_{insomnia} = 5.0$ vs $M_{better\ sleepres} = 3.0$; p<0.001 | Positive | DV: Insomnia IV: BP | $M_{SBP|insomnia} = 123$ vs $M_{SBP|better\ sleepres} = 121$; p=0.151 $M_{DBP|insomnia} = 77$ vs $M_{DBP|better\ sleepres} = 76$; p=0.441 | No association |
| Seow LS et al. (2016) [60] | Some concerns | cross-sectional (N=598; Patients with MDD; Singapore) | structured diagnostic instrument (WMH-CIDI version 3.0) | Modified Version of the CIDI 3.0 checklist | Modified Version of the CIDI 3.0 checklist of chronic medical disorders | DV: sleep disturbance IV: anxiety | OR = 3.20; 95% CI 0.40 – 27.1, p>0.05 | No association | DV: sleep disturbance IV: HTN | OR = 6.00; 95% CI 1.00 – 35.9, p=0.049 | Positive |
| Bajpai S et al. (2014) [61] | Some concerns | Retrospective (N=617; Patients with sleep disorders; USA) | ICD-9-CM | Polysomnography | ICD-9-CM | DV: depression IV: OSA | OR = 0.78; 95% CI 0.49 – 1.23, p>0.05 | No association | DV: sleep apnea IV: HTN | participants had hypertension, with a greater prevalence in the apnea group than the non-apnea group (62.7% vs 33.1%; P<.0001) | Positive |
| Chedraui P et al (2013) [62] | Some concerns | cross sectional study (N=204; Postmenopausal women; Ecuador) | The hospital anxiety and depression scale (HADS) | Athens insomnia scale (AIS) | Blood pressure measurement (≥130/85 mmHg) | DV: Insomnia IV: depression | Beta = 0.18; 95% CI 0.10 – 0.26, p<0.05 | Positive | DV: Insomnia IV: blood pressure | Correlation between insomnia and blood pressure SBP r=0.11 DBP r=0.03 | No association |
| Lu K et al. (2015) [51] | Some concerns | Cross-sectional (N=4144; adults; Mexico) | Patient Health Questionnaire-9 (PHQ-9 scale) | PSQI | Calibrated standard mercury sphygmomanometer (SBP≥140mm Hg and/or DBP≥90mm Hg) | DV: Sleep quality IV: depression | *Higher mean depression score was significantly associated with poor sleep quality (p<0.01)* | Positive | DV: Hypertension IV: Sleep quality | OR= 2.30; 95%CI: 1.68–3.17, p<0.01 | Positive |
| Gharsalli H, et al. (2022) [53] | Some concerns | Cross-sectional study (N=80; adults with OSA; Tunisia) | Hospital Anxiety and Depression Scale (HADS) | polygraphy (Embletta, Cidelec) | Self-reported (dichotomic (Yes/No)) | DV: AHI IV: depression | r=0.095, p>0.05 | No association | DV: HTN IV: AHI | HTN was not related to AHI categories, p=0.178 | No association |

DV: dependent variable, IV: independent variable, OR: Odds Ratio, CI: confidence interval, HTN: hypertension, PSQI: Pittsburgh Sleep Quality Index.

**Table 4. Description and results of the studies demonstrating two types of association, the association between sleep problems and HTN and the association between psychiatric disorders and HTN.**

| Author (year) | Number of authors | Study design (sample size, population type, location) | Measurement tools | | | Outcome results | | | | | |
|---|---|---|---|---|---|---|---|---|---|---|---|
| | | | Psychiatric disorders | Sleep problems | Hypertension | Association between sleep problems and hypertension [S-H] | | | Association between hypertension and psychiatric disorders [P-H] | | |
| | | | | | | Variables | results | Type of association | Variables | results | Type of association |
| Balog P et al. (2017) [65] | Low | Longitudinal study (N=4254; Adults; Hungary) | shortened Hungarian version [BDI-S] | four items (three for vital exhaustion (MQ-S scale) and depression (BDI-S)) | Self-reported question | DV: incidence of HTN IV: Sleep problems | OR = 1.16; 95% CI 1.00 – 1.33, p=0.044 | Positive association | DV: incidence of HTN IV: depressive symptomatology | OR = 0.95; 95% CI 0.82 – 1.11, p=0.111 | No association |
| Hein M et al. (2019) [66] | Low | Cross-sectional study (N=703; Individuals with major depression; Belgium) | Beck Depression Inventory (BDI) | polysomnography | self-reports either a physician-diagnosis or taking anti-hypertensive medication | DV: HBP IV: Objective insomnia | OR = 2.19; 95% CI 1.26 – 3.81, p=0.011 | Positive association | DV: HBP IV: depression severity | OR = 0.89; 95% CI 0.62 – 1.28, p=0.525 | No association |
| Cheng X et al (2022) [63] | Some concerns | Longitudinal study (N=261,267; patients with CVD; UK) | self-reported, antidepressant use, and depression-related hospitalization records | Self-reported sleep duration | self-report, medication, operation and electronic health records | DV: HTN IV: Sleep duration | HRa = 1.13; 95%CI 1.10 - 1.16; P<0.001 | Positive association | DV: HTN IV: depression | HRa=1.29; 95% CI 1.24 – 1.33, p<0.001 | Positive association |
| Bathgate CJ et al. (2016) [64] | Some concerns | Cross-sectional study (N=255; adults; USA) | Inventory for Diagnosing Depression | Polysomnography | Self-reported | DV: HTN IV: insomnia with objectively short sleep <6h | OR = 3.59; 95% CI 1.58 - 8.17; P=0.002 | Positive association | DV: HTN IV: depression | OR = 1.04; 95% CI 1.01 – 1.06, p<0.016 | Positive association |

DV: dependent variable; IV: independent variable; HTN: hypertension; OR: odds ratio, CI: Confidence interval; HRa: adjusted Hazard ration, RR: risk ratio; HBP: high blood pressure; CCB: Calcium channel blockers; CVD: cardiovascular disease; MS: metabolic syndrome.

**Table 5. Description and results of the studies demonstrating one type of association, the association between sleep problems and hypertension.**

| Author (year) | Risk of Bias | Study design (sample size, population type, location) | Measurement tools | | Outcome results | | |
| | | | Sleep problems | Hypertension | Variables | Results | Types of association |
|---|---|---|---|---|---|---|---|
| | | | | | **Association between sleep problems and hypertension [S-H]** | | |
| Chami HA et al. (2023) [67] | Low | Prospective observational study (N=501; adults; Lebanon) | Berlin Questionnaire | Self report HTN | DV: sleep apnea IV: Hypertension | OR = 4.33, 95% CI: 2.28–8.22, p<0.001 | Positive |
| Clark AJ et al. (2016) [68] | Low | Longitudinal cohort (N=70049; adults; Denmark, Finland, London) | Jenkins Sleep Problem Scale | physician diagnosis | DV: Hypertension IV: Disturbed sleep | HR=1.22, 95% CI: 1.04–1.44, p<0.05 | Positive |
| Li X et al. (2021) [69] | Low | Prospective cohort (N=11623; adults; USA) | Women's Health Initiative Insomnia Rating Scale (WHIIRS) | hypertension was defined as a SBP>140 mm Hg, a DBP>90 mm Hg, or the receipt of antihypertensive medication | DV: Hypertension IV: Insomnia | OR= 1.37; 95% CI: 1.11–1.69, p<0.01 | Positive |
| Schwartz J et al. (2013) [70] | Low | Cross sectional (N=126; Adults aged >55 years; USA) | Objective sleep/wake activity was measured with the Actiwatch-Light | Resting blood pressure was measured three times using non-invasive Microlife BP monitor | DV: Hypertension IV: nighttime sleep duration | OR= 0.97; 95%CI: 0.62–1.52, p>0.05 | No association |
| Lyons R et al. (2022) [71] | Low | Subset of a randomized controlled trial (N=48; veterans; USA) | Sleep monitoring system | Self-reported question | DV: objective OSA status IV: Hypertension | Hypertension was not statistically significant with OSA status (x2 = 3.00, p=0.08) | No association |

DV: dependent variable; IV: independent variable; OR: odds ratio, CI: Confidence interval; HR: Hazard ration, OSA: Obstructive sleep apnea.

Table 6. Description and results of the studies demonstrating one type of association, the association between sleep problems and psychiatric disorders.

| Author (year) | Risk of Bias | Study design (sample size, population type, location) | Measurement tools | | Outcome results Association between sleep problems and psychiatric disorders [S-P] | | Types of association |
|---|---|---|---|---|---|---|---|
| | | | Psychiatric disorders | Sleep problems | Variables | Results | |
| Shen J et al. (2020) [72] | Low | Cohort study (N=27911; adults; China) | GAD-2 | PSQI | DV: Anxiety IV: Poor sleep quality | OR: 3.85, 95% CI: 3.42–4.33, p<0.05 | Positive association |
| Jo S et al. (2020) [73] | Low | Cross-sectional (N=126; persons with epilepsy; South Korea) | PHQ-9 | ESS score | DV: ESS score IV: depression | Beta=3.11, p<0.001 | Positive association |
| Poole L et al. (2019) [74] | Low | Longitudinal Study (N=7395; Elderly, England) | Centre for Epidemiological Studies Depression scale (CES-D) | three questions (Jenkins Sleep Problems Scale) | DV: depression IV: sleep problems | r=0.35, p<0.001 | Positive association |
| El-Solh AA et al. (2022) [75] | low | Retrospective Study (N=19080; Veterans, USA) | Self-reported (Yes/No) | ICD-9; ICD-10 | DV: PTSD+Insomnia IV: depression | Depression was more frequent in patients with PTSD plus insomnia, than either PTSD or insomnia alone | Positive association |
| Morikawa et al. (2013) [76] | low | cross-sectional study (N=3796; elderly; Japan) | 15-item Geriatric Depression Scale (GDS-15) | PSQI | DV: depression IV: Sleep disturbance | OR = 2.22; 95% CI 1.83 – 2.70, p<0.001 | Positive association |
| Manzar DI (2020) [77] | Some concerns | cross sectional study (N=484; university students; Ethiopia) | Generalized anxiety disorder – 7 scale | Brief insomnia tool that comprise four items | DV: insomnia IV: anxiety | OR = 2.86; 95% CI 1.10 – 7.49, p<0.05 | Positive association |
| Palagini L et al. (2016) [80] | Some concerns | Cross-sectional (N=330; Patients with hypertension; Italy) | Perceived Stress Scale (PSS) | Insomnia severity index (ISI) | DV: Perceived stress IV: Insomnia | Beta = -0.09, p=0.251 | No association |
| Blanc J et al. (2021) [81] | Some concerns | Cross-sectional (N=700; females; USA) | Stress Index Scale | Sleep disorder questionnaire | DV: sleep disturbance IV: stress | r=0.057; p>0.05 | No association |
| Vallières A et al. (2021) [78] | High | Cross-sectional (N=200; adults; Canada) | MINI | Structured insomnia interview (SII) | DV: sleep disorders IV: number of psychiatric conditions | (26.5%) with at least one sleep disorder also had two or more psychiatric condition | Positive association |
| Yaméogo NV et al. (2015) [79] | High | Cross-sectional study (N=414; Hypertensive participants; Burkina Faso) | Hospital Anxiety and Depression Scale | European Sleep Center questionnaire | DV: sleep apnea IV: depression | Depression was more common in those with sleep apnea syndrome (p=0.004) | Positive association |

DV: dependent variable; IV: independent variable; OR: odds ratio, CI: Confidence interval.

### Studies demonstrating two sets of two associations

Studies demonstrating two types of association are presented in Tables 2–4.

### Association between sleep problems and psychiatric disorders [S-P] and between psychiatric disorders and hypertension [P-H]

Four studies [41–44] conducted on different populations examined the association between sleep problems and psychiatric disorders [S-P] and between psychiatric disorders and hypertension [P-H] (Table 2).

Both types of association were positive in two cross-sectional studies [41,43], with some risk of bias. One involving adult women found a positive association between anxiety and self-reported sleep problems and between self-reported anxiety and hypertension [41]. Another study conducted among 40 patients with congestive heart failure (CHF) reported a positive correlation between depression severity, as measured by the Beck Depression Inventory (BDI), and insomnia diagnosed by PSQI-IV [43]. This same study found a positive relationship between depression and hypertension, with blood pressure data retrieved from medical records [43].

One study conducted among 846 patients with hypertension showed a positive association between depression, measured by the PHQ-9 tool, and self-reported sleep duration [42]. However, this study, which had methodological concerns, found no association between depressive symptoms and hypertension measured by a clinical device [42]. Lastly, a longitudinal study with minimal methodological issues involving 73 patients with Parkinson's disease found no association in either relationship [44].

### Association between sleep problems and psychiatric disorders [S-P] and between sleep problems and hypertension [S-H]

Eighteen studies examined the association between sleep problems and psychiatric disorders [S-P] and between sleep problems and hypertension [S-H] (Table 3). Sleep problems were assessed using various tools, including self-reported questionnaires, diagnostic methods (polysomnography, polygraphy, ICD-9, ICD-10, and DSM-IV), and validated scales such as the Leeds Sleep Evaluation Questionnaire, PSQI, and Athens Insomnia Scale. Psychiatric disorders were measured using tools such as self-reported questionnaires, MINI diagnosis, the 12-item General Health Questionnaire, DSM-IV, ICD-9, ICD-10, Hospital Anxiety and Depression Scale (HADS), BDI-II, and PHQ-9. Hypertension was measured through self-reported diagnoses, ICD-9 and ICD-10 codes, and clinical devices.

Seven studies [45–51] reported a positive association for both S-P and S-H relationships, while two studies [52,53] found no such association. Of these seven studies, four were longitudinal and involved patients with acute coronary syndrome (ACS), psychiatric patients, older adults, and individuals aged 16 and older [45,48–50]. The remaining three studies were cross-sectional and included adults, older adults, and individuals with hypertension [46,47,51]. Moreover, nine studies found a positive correlation for one type of relationship but no correlation for the other type [54–62].

Thirteen studies were considered to have a low risk of bias [45–50,52,54–59], while five presented methodological concerns [51,53,60–62]. For studies examining the [S-P] association, the assessed variables were depressive disorders on one hand and sleep duration, sleep quality, insomnia, and sleep problems on the other. Thirteen of the 18 studies found a positive association across various populations [45–51,54–57,59,62], while five reported non-significant associations between the [S-P] relationship [52,53,58,60,61], although two had high-quality methods and large sample sizes [52,58].

In studies exploring the [S-H] relationship, sleep variables included sleep duration, quality, disturbances, insomnia and sleep problems, and sleep apnea, while hypertension was measured by blood pressure devices. Ten studies reported a positive association [45–51,58,60,61], while eight did not [52–57,59,62]. Both groups involved studies with large sample sizes and reliable methodologies (Table 3).

## Associations between sleep problems and hypertension [S-H] and between psychiatric disorders and hypertension [P-H]

The associations between sleep problems and hypertension [S-H] and between psychiatric disorders and hypertension [P-H] were examined in four studies, as detailed in Table 4. Two studies [63,64] found a positive association for both types of relationships, while no association for [P-H] was observed in the other two studies [65,66]. The two studies with positive associations had some methodological concerns [63,64]; one was cross-sectional, involving 255 adults [64], and the other was a cohort study conducted among 261,267 patients with cardiovascular disease [63]. In these studies, self-reported hypertension was the dependent variable in both associations. The two studies with fewer methodological concerns found a positive association between sleep problems and hypertension [S-H] but no association between hypertension and psychiatric disorders [P-H] [65,66] (Table 4).

## Studies demonstrating one set of two-association

Studies investigating one association between any two of the three disorders are presented in Tables 5 and 6. Psychiatric disorders were measured using tools such as self-reported diagnoses, GAD-2, GAD-7, PHQ-9, MINI, Perceived Stress Scale (PSS), Stress Index Scale, CES-D, HADS, and the Geriatric Depression Scale (GDS). Sleep problems were assessed through diagnostic interviews, ICD-9, ICD-10, PSQI, ESS score, ISI, Sleep Disorder Questionnaire, JSS, European Sleep Center questionnaire, and the brief insomnia tool.

As shown in Table 5, the association between sleep problems and hypertension [S-H] was found in five studies. A positive S-P association was noted in three longitudinal studies assessing hypertension and sleep apnea among large adult samples [67–69]. In contrast, two studies [70,71] reported no significant association between sleep problems and hypertension. Despite having small sample sizes, these studies were of high quality: one was conducted among 48 veterans, a subset of a randomized controlled trial with appropriate measures [71], and the other among 126 adults over the age of 55 using a cross-sectional design with self-reported hypertension [70].

Table 6 presents the association between sleep problems and psychiatric disorders [S-P] observed in ten studies. Eight studies reported a positive association [72–79], including three cohort studies [72,74,75] and five cross-sectional studies [73,76–79], most of which had large sample sizes and reliable methodologies. Conversely, the two studies [80,81] that did not find significant associations between stress and insomnia or sleep disturbances had some methodological flaws: one focused on patients with hypertension [80] and the other on female participants [81].

## Summary of the results

Overall, among twelve studies related to the association of hypertension with psychiatric disorders [P-H], five studies (41.67%) reported significant positive associations [40,41,43,63,64], while seven (58.33%) showed non-significant results [32,33,39,42,44,65,66]. We note that all studies with positive [P-H] associations were of moderate quality (none were of good quality) [40,41,43,63,64], while five of the studies with non-significant results were of high-quality [32,39,44,65,66], with acceptable or even large sample sizes. Nevertheless, three of the seven studies with non-significant associations involved specific sub-populations, such as those with major depression [66], hypertension [42], or Parkinson's disease [44]. In summary, while most studies (60%) reported a positive [P-H] association, caution is warranted as some were not statistically significant.

Out of the 31 studies examining the association between sleep problems and hypertension [S-H], 20 (64.51%) found positive associations [32,33,40,45–51,58,60,61,63–69], while 11 (34.37%) reported non-significant results [39,52–57,59,62,70,71]. Among the studies showing positive associations, the majority were of high (n = 13) [32,45–50,58,65–69] to moderate (n = 7) quality [33,40,51,60,61,63,64]. No distinct pattern emerged among studies with non-significant results, as some had a low risk of bias and involved large adult samples, while others were of lower quality

or focused on specific populations, such as postmenopausal women or veterans. Consequently, while a probable positive [S-H] association can be inferred, caution is warranted in drawing firm conclusions.

As for the association of sleep problem-psychiatric disorder [S-P], 28 out of 36 studies [32,33,39–43,45–51,54–57,59,62,72–79] reported positive associations (77.8%). Of these, 18 were of high quality [32,39,45–50,54–57,59,72–76], and ten were of moderate quality [33,40–43,51,62,77–79]. In contrast, eight studies found no associations (22.2%) [44,52,53,58,60,61,80,81], with five being of moderate quality [53,60,61,80,81] and three of high quality [44,52,58]. These studies had smaller sample sizes, were conducted on particular sub-populations (hypertension, depression, or Parkinson's disease), or used specific measures (e.g., sleep apnea). Overall, the evidence suggests a consistently positive [S-P] association across various populations and methodologies.

## Discussion

This systematic review is the first to explore the relationship between sleep disturbances, hypertension, and psychiatric disorders. Previous studies have examined these conditions in pairs, but none have investigated the interactions between all three conditions simultaneously. This novel approach would have provided a more comprehensive understanding of how these health issues interrelate and potentially exacerbate one another. Nevertheless, a challenge encountered in this review was the variability in definitions for sleep disturbances, hypertension, and psychiatric disorders across the included studies. This heterogeneity highlights the need for standardized criteria in future research to ensure more consistent and comparable results. Most studies showed positive (60–75%) or non-significant associations (25–40%) between these health conditions, with none reporting inverse associations. This consistent pattern suggests that these conditions are interrelated, even when statistical significance is sometimes not reached. The associations between psychiatric disorders and sleep, hypertension and sleep, and psychiatric problems and hypertension all displayed a generally positive trend, although these results should be interpreted cautiously.

The studies in this review examined pairwise associations, even those that included all three disorders simultaneously. Depressive and anxiety disorders were the most commonly represented psychiatric disorders, while insomnia was the most prevalent sleep disorder. The high prevalence of these conditions in the general population explains their frequent co-occurrence with hypertension [29]. Insomnia, which is the leading sleep disorder globally, is associated with numerous physical and mental health issues [82,83]. Similarly, anxiety and depression are the most common mental disorders worldwide, contributing significantly to disability and the total global disease burden [9]. Evidence has shown that people with hypertension often experience anxiety and depression [84–86]. Hypertension is widely recognized as a psychosomatic illness, where psychological factors, such as anxiety and depression, significantly contribute to its development, progression, and management [87]. Hypertension has both psychological and physical components, representing an interaction between the mind and body. Psychological stress can contribute to the development or worsening of hypertension, and vice versa, making it a condition influenced by both mental and physical factors. Additionally, sleep disturbances, particularly insomnia, are frequently reported in hypertensive individuals. Numerous studies have demonstrated that adults with hypertension are more prone to insomnia, with psychiatric conditions, such as depression and anxiety, further increasing the likelihood of sleep disturbances [29,88,89].

In this review, 60% of the studies found positive associations between hypertension and psychiatric events, while 40% reported non-significant results, often involving special populations. The coexistence of mental health disorders, such as anxiety and depression, with hypertension, has been previously reported [90]. Our findings align with previous systematic reviews in Ethiopia and Saudi Arabia, indicating depression prevalence in hypertensive patients of 32.43% and 57.3%, respectively [91,92]. Regarding anxiety, our results are consistent with a systematic review from 2021 showing a statistically significant positive association between anxiety and hypertension in cross-sectional (OR = 1.37, 95% CI = 1.21-1.54) and prospective (OR = 1.40, 95% CI = 1.23-1.59) studies [85]. Similarly, a meta-analysis concluded that 8 out of 13 prospective studies found an association between anxiety and an increased risk of hypertension [93]. These psychiatric

disorders may either result from hypertension, increase the likelihood of developing it, or share a common underlying cause with hypertension. While these conditions frequently occur together, their exact temporal and causal relationship is not yet fully understood and was not fully assessed. Therefore, it is essential to consider multiple factors, such as environmental factors, family stressors, financial difficulties [94], treatment side effects, especially those related to anti-psychiatric medications [95], and pathophysiological co-morbidities, when investigating the independent link between blood pressure and psychiatric disorders [96].

The majority of studies (65.62%) in this review showed a positive association between sleep disorders and hypertension. Sleep deprivation can act as a chronic stressor, disrupting homeostasis and potentially triggering heightened stress system activity, which may impair resilience and health. This perspective views hypertension as a physical response to prolonged stressful conditions, such as sleep deprivation, which may affect the neuroendocrine stress response [30]. Research has also shown that hypertensive patients often experience psychological issues, such as anxiety and depression, which are recognized risk factors for insomnia [89]. Both sleep disorders and hypertension are significant public health issues, and numerous studies have highlighted the substantial link between these conditions [29,97]. Moreover, a recent systematic review revealed that sleep disturbances are related to fluctuations in blood pressure across short-, mid-, and long-term periods. Disorders such as restless legs syndrome, shift work, insomnia, both short and long sleep durations, obstructive sleep apnea, and sleep deprivation have all been positively associated with changes in systolic and diastolic blood pressure [98]. However, due to variations in the quality of the included studies, many previous investigations, including the present review, could not fully clarify the bidirectional relationship between sleep disturbances and hypertension or distinguish between clinically diagnosed and non-clinical sleep disturbances, as done in a meta-analysis conducted in 2022 [29].

Among the studies included in this review, 27 out of 36 (75%) consistently reported positive associations between sleep disorders and mental health problems despite discrepancies in methodological quality. These results are comparable with previous systematic reviews and meta-analyses that have explored the association between sleep problems and mental health issues. For example, insomnia is associated with several mental health issues, including depression [21], anxiety, psychosis [23], and cognitive decline [99]. A recent systematic review of 52 studies has concluded that short sleep duration is an independent predictor of developing mental disorders, particularly anxiety and depression [100]. The link between sleep disturbances and depression is well documented, with shared biochemical pathways and genetic factors, as underscored by a systematic review focusing on mental and cognitive outcomes, including mental disorders and dementia [101]. This relationship may also be bidirectional, with sleep disturbances often aggravating symptom severity [102]. This bidirectional association between sleep disturbance and depression has been further emphasized, with sleep problems being no longer considered an epiphenomenon of depression but a predictive prodromal symptom [2]. People with sleep disturbances may encounter circadian rhythm disruptions and autonomic nervous system changes [103], potentially linking sleep problems with various physical and mental health conditions [104]. In this review, sleep disturbances were associated with higher odds of multimorbidity, including autonomic and mental problems.

Nevertheless, the presence of numerous high-quality studies with non-significant associations and moderate-quality studies with significantly positive associations urges caution in interpreting the results. The complex interrelationship between the variables may explain the inconsistencies in the literature regarding pathways, causal relationships, and directions of these associations. Given that this review included both cross-sectional and longitudinal designs, reciprocal interactions and reverse causation could account for some of the observed discrepancies. Several other factors may explain the variability of the results, and the non-significant associations found in some high-quality and large-sample studies. The measurements used to assess psychiatric disorders, sleep problems, and hypertension relied primarily on self-reported diagnoses, where patients' subjectivity may have influenced the findings. For instance, previous research has highlighted that sleep must be considered qualitatively and quantitatively when exploring its possible correlation with

hypertension. The subjective nature of sleep assessments could introduce bias, particularly when sleep quality is examined as an independent variable [105].

Studies suggest that subjective reports of short sleep are often less accurate than objective measures in identifying the association between hypertension and insomnia. Also, several studies found an association between objective sleep measures and increased hypertension, while subjective reports of sleep problems do not always align with objective data [25,64,106,107]. Additionally, the prevalence of hypertension could vary with the methods used to quantify hypertension (whether through objective or subjective measures). A meta-analysis found that nearly half (45%) of the 11 prospective studies it included relied solely on self-reported hypertension diagnoses or treatment, leading to the conclusion that the number of people diagnosed with hypertension would have probably increased if objective blood pressure measurements had been used [108].

Regarding psychiatric diseases, research indicates that self-reported depression may enhance the accuracy of objective diagnoses, with studies showing some alignment between self-reported depression and clinical diagnoses, although a part of the assessments might have been misclassified [109]. While objective assessments should be measurable, unbiased, and documented using diagnostic tools [110], self-reported measures remain a more cost-effective option, particularly for collecting data from large populations. Surveys and questionnaires allow researchers to quickly and efficiently reach large and diverse samples. However, subjective measures depend on human judgment, which often varies widely. Thus, the reliance on self-reporting may introduce recall and social desirability biases, leading to inaccurate diagnoses, which, in turn, could have influenced the findings of this review.

Finally, the biological plausibility of the results remains conflicting and insufficiently established. The most studied mechanism is the positive association between insomnia and hypertension through chronic inflammation and physiological hyperarousal from over-activation of the sympathetic nervous system and hypothalamic-pituitary-adrenal axis [32]. This dysregulation leads to increased cortisol secretion and hyperactivity of the sympathetic system, which exacerbates mental health conditions. Dysregulated neurotransmitters, such as serotonin and dopamine, may further contribute to the connections between insomnia, hypertension, and mental health disorders like anxiety and depression [24]. Interestingly, however, higher blood pressure has been associated with better mood and reduced emotional brain activity. This phenomenon may be explained by baroreceptor signaling, which modulates sensory and emotional processing, decreases cortical excitability, and inhibits central nervous system activity [8].

## Limitations

Several limitations should be acknowledged. The literature search did not account for the gray literature, potentially excluding eligible studies. The included studies exhibited heterogeneity in data, population characteristics, interventions, outcomes, and study designs, which complicates the synthesis and comparison of results, reducing the overall reliability of the conclusions. Also, the considerable heterogeneity across the included studies in terms of design, populations, and reported outcomes prevented us from establishing a funnel plot or a summary forest plot for the odds ratios. These plots were essential to assess publication bias and provide a pooled estimate. Moreover, the definitions and measurement tools for psychiatric disorders and sleep problems varied across studies, as did the thresholds used to determine hypertension, indicating a lack of uniformity and consensus.

Another limitation is that the search strategy excluded non-English language studies, potentially affecting the generalizability of the findings. Additionally, subjectivity during study selection and data extraction could have led to selection bias. The results of the included studies may also be skewed since studies with positive outcomes are more likely to be published than those yielding negative or ambiguous results. Regarding the biological focus, we did not specifically target biological factors in our search strategy, which is why they were not considered in this review. A quantitative meta-analysis of homogeneous studies and an assessment of publication bias are recommended to address these issues.

## Public health implications and recommendations

The co-occurrence of common mental disorders, such as anxiety and depression, with sleep disturbances and hypertension represents a growing global public health concern. These highly prevalent conditions are closely interrelated, often reinforcing one another and contributing to a cycle of poor health, increased healthcare costs, and reduced quality of life [111]. Addressing them in isolation may limit the effectiveness of public health efforts. Therefore, reducing their collective burden requires early screening, integrated care models, and comprehensive prevention strategies aimed at improving population health outcomes.

In addition, understanding the relationship between these conditions could potentially improve clinical management of these comorbidities and significantly enhance patient quality of life. The primary challenge lies in determining a common pathway that elucidates the observed triangular relationship between these conditions. The findings of this review could not yet achieve this goal, and efforts are still necessary to clarify these connections. Global longitudinal registries with valid definitions and standard tools for sleep disorders, psychological conditions, and hypertension measurements are recommended. This approach would facilitate the standardization of research methods and enable simultaneous measurement of these factors. Special attention should be given to specific subgroups such as women, older individuals, and veterans. Adapted tools should be developed for these populations, and stratified analyses should be conducted to account for their specificities.

Among the three associations examined in this review, the association between hypertension and psychiatric disorders [P-H] was the least studied and yielded mixed results despite a trend toward positivity. Importantly, many of the non-significant studies were of higher quality but focused on special populations. More high-quality studies using globally accepted standard measures are recommended, especially for general populations.

The association between sleep and hypertension, despite being more widely studied, also produced mixed results. However, the evidence tends toward a positive association, confirmed by a positive biological plausibility. More studies using standard measures are needed, and meta-analyses of the most homogeneous studies are suggested to resolve the uncertainty in this field.

In contrast, the relationship between sleep disorders and psychiatric conditions [S-P] produced the most consistent positive results. A meta-analysis of the most homogeneous findings could provide a definitive conclusion to this association.

## Conclusion

This systematic review provides a comprehensive synthesis of the existing evidence regarding the association between sleep problems, psychiatric disorders, and hypertension, demonstrating that the majority of the studies reported positive or non-significant associations, with no study reporting inverse associations between these three conditions. However, the substantial heterogeneity in study methodologies requires a cautious interpretation of the results. The measurements used to assess these disorders varied, including self-reports, clinical diagnoses, validated scales, and clinical devices. The bi-directional association between the three illnesses emphasizes the need for standard definitions and clearer screening protocols, where diagnosing one disorder prompts evaluation for the others and highlights the potential for simultaneous treatment. Future studies should focus on uncovering the biological mechanisms underlying these conditions and larger studies of better quality. Research should concentrate on determining the factors that contribute to the triangle relationship that exists between hypertension, sleep disorders, and mental illnesses. Understanding the processes behind this complex relationship would be greatly enhanced by looking at the bidirectional interactions between these conditions as well as any mediating and moderating factors. Continued efforts in this field are crucial for advancing scientific understanding, optimizing patient outcomes, and informing best practices in clinical care. Furthermore, based on the biological pathways found, future studies are warranted to investigate possible therapeutic targets and consider how multidisciplinary methods could improve treatment approaches.

## Supporting information

**S1 Prisma Checklist. PRISMA checklist used for reporting the systematic review.**
(DOCX)

**S1 File. Risk of Bias Assessment Tool (RoBINS-E).** Risk of Bias evaluation for included studies using the RoBINS-E tool.
(PDF)

**S1 Data. List of Included Studies: Detailed list of all studies identified in the systematic review.**
(XLS)

## Acknowledgments

The authors extend their sincere gratitude to Marie-Therese Mitri, Senior Director of Access Services, Learning & Research Support at LAU University Libraries, for her invaluable assistance in acquiring the necessary articles, and Carla Hanna for her essential support in developing this review.

## Author contributions

**Conceptualization:** Pascale Salameh, Sola Aoun Bahous.

**Data curation:** Chadia Haddad, Hala Sacre.

**Formal analysis:** Chadia Haddad, Samah Tawil.

**Investigation:** Sola Aoun Bahous.

**Methodology:** Chadia Haddad, Samah Tawil.

**Supervision:** Pascale Salameh, Sola Aoun Bahous.

**Validation:** Hala Sacre.

**Writing – original draft:** Chadia Haddad, Samah Tawil.

**Writing – review & editing:** Chadia Haddad, Hala Sacre, Samah Tawil, Pascale Salameh, Sola Aoun Bahous.

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
