## [Decision Letter · Decision Letter 0]

21 Jan 2025

PGPH-D-24-02268The Triangulation of Multimorbidity: A Systematic Review of Sleep Disturbances, Hypertension, and Psychiatric IllnessesPLOS Global Public Health

Dear Dr. Haddad,

Thank you for submitting your manuscript to PLOS Global Public Health. After careful consideration, we have decided that your manuscript does not meet our criteria for publication and must therefore be rejected.

We are sorry that we cannot be more positive on this occasion. We very much appreciate your wish to present your work in one of PLOS's Open Access publications. Thank you for your support, and we hope that you will consider PLOS Global Public Health for other submissions in the future.

Yours sincerely,

Feten Fekih-Romdhane

Academic Editor

Additional Editor Comments (if provided):

Reviewers' comments:

Reviewer's Responses to Questions

**Comments to the Author**

1. Does this manuscript meet PLOS Global Public Health’s publication criteria ? Is the manuscript technically sound, and do the data support the conclusions? The manuscript must describe methodologically and ethically rigorous research with conclusions that are appropriately drawn based on the data presented.

Reviewer #1: Yes

Reviewer #2: No

Reviewer #3: Yes

2. Has the statistical analysis been performed appropriately and rigorously?

Reviewer #1: Yes

Reviewer #2: N/A

Reviewer #3: Yes

3. Have the authors made all data underlying the findings in their manuscript fully available (please refer to the Data Availability Statement at the start of the manuscript PDF file)?

Reviewer #1: Yes

Reviewer #2: Yes

Reviewer #3: Yes

4. Is the manuscript presented in an intelligible fashion and written in standard English?

Reviewer #1: Yes

Reviewer #2: Yes

Reviewer #3: Yes

5. Review Comments to the Author

Reviewer #1: The abstract lacks depth in explaining the mechanisms behind the observed associations, leaving the findings superficial and not actionable.

The methods section is vague, failing to detail how the heterogeneity of study definitions and methods was addressed, which weakens the reliability of the conclusions.

The conclusion is overly cautious and does not provide clear implications or directions for future research, making the study's contribution to the field unclear.

Reviewer #2: The conceptual framework of the study raises some concerns. Insomnia is classified as a psychiatric disorder in both the ICD-11 and DSM-5 and symptom of some disorders, making its relationship with other mental disorders, particularly anxiety and depression as highlighted in the included articles, an anticipated finding. Furthermore, insomnia frequently manifests as a symptom of these mental health conditions. This underscores the importance of employing the term "disorders" rather than "illnesses" in discussions of mental health, given that mental disorders are significantly influenced by cultural factors and rely on professional consensus rather than universally defined signs or symptoms.

There is also an issue with the discussion of genetic and neurotransmitter pathways. While these pathways are undoubtedly relevant, they are common across many mental disorders and cannot be uniquely associated with insomnia.

Regarding the "holistic approach" described in the paper, this characterization appears inconsistent with its predominantly biological perspective. Mental disorders are increasingly understood through a biopsychosocial lens, and incorporating a broader perspective might enhance the paper's relevance and impact.

The rationale for prioritizing the specific association between insomnia, hypertension, and anxiety/depression is unclear. This might explain why only three articles directly examine this relationship in the context of the study's aim.

The results presented do not seem to provide consistent support for the hypothesis. Although a “positive trend” is mentioned, the findings appear insufficient to substantiate the conclusions drawn in the discussion. Furthermore, the biological focus of the review might be more suited to a journal specializing in biological research. Hypertension, mental disorders, and sleep disorders are deeply influenced by social and psychological factors. For a journal with a focus on global public health, the study could offer greater value by providing insights that contribute meaningfully to global mental health, rather than predominantly reflecting a Western biological perspective.

Reviewer #3: Thank you for giving chance to review this manuscript.

Overall this study is excellent and interesting, very organized manuscript. This study had great importance in clinical area for medical professionals to understand the link between mental and medical illness in general for better screening and treatment. For the future researchers it gives great attention to focus on the longitudinal Couse –effect relationship between mental health problems and hypertension.

1.Title of the study

Your study report that there is an association between psychiatric disorders and sleep, hypertension and sleep, and psychiatric problems and hypertension all displayed a generally positive trend.

But The DSM-5 classification of sleep-wake disorders is intended for use by mental health and general medical clinicians (those caring for adult, geriatric, and pediatric individuals). Sleep-wake disorders encompass 10 disorders or disorder groups: insomnia disorder, hypersomnolence disorder, narcolepsy, breathing-related sleep disorders, circadian rhythm sleepwake disorders, non–rapid eye movement (NREM) sleep arousal disorders, nightmare disorder, rapid eye movement (REM) sleep behavior disorder, restless legs syndrome, and substance/medication-induced sleep disorder. Individuals with these disorders typically present with sleep-wake complaints of dissatisfaction regarding the quality, timing, and amount of sleep. Resulting daytime distress and impairment are core features shared by all of these sleep-wake disorders

•How it could be assessed the relationship between depression /anxiety and sleep disorder. Since sleep disorder by itself part of mental disorder and sleep disturbance one of the common symptoms in depression and anxiety. Are you assessed primary sleep disorder mentioned above or secondary sleep disorder due to depression and anxiety? Make it clear.

2.Abstract

Conclusion should be smart. Put recommendation for the result of the your study.

3.Introduction

•You have Sleep disorders, including insomnia and obstructive sleep apnea (OSA). Please describe all sleep disorder classification in DSM-IV or DSM-5 mentioned above.

•Insomnia, hypertension, and mental health are linked through complex, bidirectional relationships ( add Reference for this statement)

4.Method

•This study defines Psychiatric illnesses: (e.g., anxiety, depression, psychosis, obsessive-compulsive disorder, or PTSD). Why other psychiatric disorders excluded example bipolar disorder, sexual disorders, somatic symptom disorders etc.... it is better to specify the reason.

5.Discussion

•What does it mean Hypertension is widely recognized as a psychosomatic illness? Somatic symptom disorder in psychiatry context considered as complain of many different medical symptoms without any positive physical and laboratory finding.

•If you considered psycho-somatic (psycho for depression and anxiety and somatic for hypertension), make it clear this study didn’t describe somatic symptom related disorders.

•What does it mean Psychological conditions/factors, such as depression and anxiety? Depression and anxiety are psychiatric condition but not psychological. Psychological and psychiatric problems have bilateral association however depression and anxiety are psychiatric condition; it is better to update this study in this way.

6.Strengths and Limitations

You can describe only your study limitation. What you did throughout in whole manuscript or study is the strength of the study. It is better to put statement for limitation of the study only.

7.Recommendation

Recommend the future studies to focus on factors contributing triangular association of psychiatric disorders, sleep and hypertension.

6. PLOS authors have the option to publish the peer review history of their article (what does this mean? ). If published, this will include your full peer review and any attached files.

**Do you want your identity to be public for this peer review?** For information about this choice, including consent withdrawal, please see our Privacy Policy .

Reviewer #1: No

Reviewer #2: No

Reviewer #3: **Yes: ** Tamene Berhanu Alaho

---

## [Editor Report · Decision Letter 1]

25 Jun 2025

PGPH-D-24-02268R1

The Triangulation of Multimorbidity: A Systematic Review of Primary Sleep Disorders, Hypertension, and Psychiatric Disorders

Dear Dr. Haddad,

Thank you for submitting your manuscript to PLOS Global Public Health. After careful consideration, we feel that it has merit but does not fully meet PLOS Global Public Health’s publication criteria as it currently stands. Therefore, we invite you to submit a revised version of the manuscript that addresses the points raised during the review process.

We look forward to receiving your revised manuscript.

Kind regards,

Nancy Angeline Gnanaselvam

Academic Editor

Additional Editor Comments (if provided):

Figure 1 is pixelated

I appreciate the authors for conducting the meticulous systematic review. The article needs further revision to proceed to next stage.

Authors need to elaborate in introduction and discussion on the relevance of this article and the scope of PGPH Journal. The public health implications of insomnia, hypertension and Common mental illnesses needs to be mentioned.

Funnel plot can be added to describe any publication bias

Odds ratios cab be presented in forest plot format
---

## [Editor Report · Decision Letter 2]

31 Jul 2025

PGPH-D-24-02268R2

The Triangulation of Multimorbidity: A Systematic Review of Primary Sleep Disorders, Hypertension, and Psychiatric Disorders

Dear Dr. Haddad,

Thank you for submitting your manuscript to PLOS Global Public Health. After careful consideration, we feel that it has merit but does not fully meet PLOS Global Public Health’s publication criteria as it currently stands. Therefore, we invite you to submit a revised version of the manuscript that addresses the points raised during the review process.

We look forward to receiving your revised manuscript.

Kind regards,

Nancy Angeline Gnanaselvam

Academic Editor

Journal Requirements:

Additional Editor Comments (if provided):

Thank you for addressing all queries promptly. Kindly note that you have mentioned all studies in the table and yet the citation list is too long. Please be succinct.
---

## [Editor Report · Decision Letter 3]

2 Sep 2025

The Triangulation of Multimorbidity: A Systematic Review of Primary Sleep Disorders, Hypertension, and Psychiatric Disorders

PGPH-D-24-02268R3

Dear Dr. Haddad,

We are pleased to inform you that your manuscript 'The Triangulation of Multimorbidity: A Systematic Review of Primary Sleep Disorders, Hypertension, and Psychiatric Disorders' has been provisionally accepted for publication in PLOS Global Public Health.

Best regards,

Nancy Angeline Gnanaselvam

Academic Editor
